# Evolutionary insights into 3D genome organization and epigenetic landscape of *Vigna mungo*

Alim Junaid, Baljinder Singh, Sabhyata Bhatia

Eukaryotic genomes show an intricate three-dimensional (3D) organization within the nucleus that regulates multiple biological processes including gene expression. Contrary to animals, understanding of 3D genome organization in plants remains at a nascent stage. Here, we investigate the evolution of 3D chromatin architecture in legumes. By using cutting-edge PacBio, Illumina, and Hi-C contact reads, we report a gap-free, chromosome-scale reference genome assembly of *Vigna mungo*, an important minor legume cultivated in Southeast Asia. We spatially resolved *V. mungo* chromosomes into euchromatic, transcriptionally active A compartment and heterochromatic, transcriptionally-dormant B compartment. We report the presence of TAD-like-regions throughout the diagonal of the HiC matrix that resembled transcriptional quiescent centers based on their genomic and epigenomic features. We observed high syntenic breakpoints but also high coverage of syntenic sequences and conserved blocks in boundary regions than in the TAD-like region domains. Our findings present unprecedented evolutionary insights into spatial 3D genome organization and epigenetic patterns and their interaction within the *V. mungo* genome. This will aid future genomics and epigenomics research and breeding programs of *V. mungo*.

## Introduction

The non-random packaging of chromatin within the nucleus is a universal feature of eukaryotic genomes. The three-dimensional (3D) spatial organization of chromatin could be partitioned at different levels based on the interaction frequency between two given loci in the genome. Advances in sequencing technologies have led to the identification of organization of individual chromosomes of the eukaryotic genome into two spatially discrete compartments—A and B, at megabase resolution, which is driven by phase separation of chromatin bearing similar modifications (Lieberman-Aiden et al, 2009). The B compartment is localized close to the nuclear lamina and enriched in transcriptionally inactive heterochromatin, whereas the A compartment is enriched in transcriptionally active euchromatin and localized in the internal regions of the nucleus (Lieberman-Aiden et al, 2009; Bickmore & van Steensel, 2013; Rao et al, 2014). A more granular analysis of chromatin contact maps at kilobase resolution led to the identification of topologically associated domains (TADs) in the eukaryotic genome (Dixon et al, 2012). TADs are self-interacting contiguous structural units of sub-chromosomal scale in the 3D hierarchical organization of eukaryotic genomes (Jin et al, 2013; Rao et al, 2014). In mammals, flanking regions of TADs are enriched in CCCTC-binding factor (CTCF) and structural maintenance of chromosomal complex proteins where they are associated with the formation and maintenance of TADs by loop extrusion mechanism (Dixon et al, 2012; Sexton et al, 2012; Rao et al, 2014). TADs are proposed to form important functional elements in the genome and mediate several crucial functions—(1) regulating gene expression by harboring regulatory elements with their target genes (Dixon et al, 2016), (2) facilitating enhancer elements to mediate "non-specific" co-regulation of genes residing within the TADs (Symmons et al, 2014; Dixon et al, 2016), and (3) insulating genes from potential detrimental effects of regulatory elements located in the vicinity of TADs (Dixon et al, 2012). Consequently, altering TADs structure results in changes in gene expression (Drubin et al, 1986; Dowen et al, 2014; de Wit et al, 2015; Gomez-Marin et al, 2015; Guo et al, 2015; Lupianez et al, 2015; Narendra et al, 2015; Sanborn et al, 2015; Symmons et al, 2016; Paliou et al, 2019). Recent studies have identified some degree of conservation of TADs in genomes of closely related species further emphasizing their functional relevance—for example, TADs exhibit at least 30–40% of conservation in the genomes of *Drosophila melanogaster* and *D. pseudoobscura* and about 43% between humans and their closest relative chimpanzees. Unlike animals, understanding of 3D genome architecture in plants is at nascent stages and needs to be investigated rigorously. This is because decoding spatial 3D organization of plant genomes will provide an excellent opportunity to study comprehensive long-range interaction between expression quantitative trait loci (eQTL) and regulatory elements that could have functional implications in modern breeding. Unlike animals, CTCF is not found in plants. Therefore, the mechanism of TAD formation in plant genomes still remains debatable. Recent Hi-C–based studies of spatial chromatin conformation in rice, *Brassica*, cotton, pepper, *G.*

National Institute of Plant Genome Research, Aruna Asaf Ali Marg, New Delhi, India

Correspondence: sabhyatabhatia@nipgr.ac.in

*max*, and *P. vulgaris* have reported the presence of TAD-like structures (Liu et al, 2017; Dong et al, 2018; Wang et al, 2018, 2021a, 2021b; Liao et al, 2022). However, such investigations have been sparse in nutritionally enriched minor pulse crops which are mainly cultivated in developing countries.

The genus *Vigna* belongs to the *Fabacae* family and is subdivided into five subgenera—*Ceratotropis*, *Haydonia*, *Lasiocarpa*, *Plectotropis*, and *Vigna*. The domesticated species belong to the three subgenera—Ceratotropis, Plectrotropis, and Vigna. *Vigna* species such as black gram (*Vigna mungo* [L.] Hepper), adzuki bean (*Vigna angularis* [Willd.] Ohwi & Ohashi) mung bean (*Vigna radiata* [L.] R. Wilczek) belong to *Ceratotropis* subgenus, whereas cowpea (*Vigna unguiculata* [L.] Walp) belongs to *Vigna* subgenus (Takahashi et al, 2016). These are important minor legume crops and good sources of dietary proteins in the Southeast Asian region. Black gram, also commonly known as urdbean in India, is a diploid, self-pollinating crop with an estimated genome size of ~574 Mb (Arumuganathan & Earle, 1991). Reference genome sequences of several commercial *Vigna* species such as adzuki bean, cowpea, mung bean, and asparagus bean are already available (Kang et al, 2015; Lonardi et al, 2019; Xia et al, 2019; Ha et al, 2021). De novo-based chromosomal or draft assembly of *V. mungo* has been previously reported using Illumina, long-range Chicago 10X genomics and Hi-C reads. However, these assemblies contain large sequencing gaps and lack completeness (Pootakham et al, 2021). Gaps in a genome assembly can largely impair genome-scale studies, for example, many sequence alignment tools may have reduced performance because of the presence of gaps in query sequences (Chen & Tompa, 2010; Song et al, 2018). Gaps can also result in faulty genome annotations in intraspecific genome comparisons (Bickhart & Liu, 2014; Song et al, 2019). Moreover, availability of complete and accurate genome information is essential to understand the 3D organization of the genome and its evolution and diversity, genome annotation, epigenetic regulation, and identifying structural variants associated with agronomically important traits. In this study, we report a nearly complete, improved contiguous chromosome-level genome assembly of *V. mungo* with a gap of only 14 kb, by integrating SMRT long read, Hi-C, and Illumina pair-end sequencing technologies. We then use the Hi-C contact matrix to unveil the hierarchical chromatin organization and investigate its regulatory role in gene transcription. We further show trends of evolutionary conservation of spatial organization of 3D chromatin structure in legumes. In addition, using BS-seq and RNA-seq technologies, we investigate the relationship between epigenetic patterns and 3D genome organization and gene expression. The genome sequence of *V. mungo* reported here with "filled-in" sequencing gaps and increased contiguity along with the epigenome and spatial 3D genome architecture will serve as a valuable resource for genomics and epigenomics research and future breeding programs of *V. mungo*.

## Results

### De novo assembly and annotation of the *V. mungo* genome

A de novo assembly of *V. mungo* has been reported previously; however, it contains a large sequencing gap of 34 Mb (Pootakham et al, 2021). To generate a high-quality, chromosome-level assembly of *V. mungo* genome, we applied a multitiered scaffolding approach of PacBio SMRT long-reads, Illumina short-reads, and Hi-C sequencing technologies. We generated a total of ~56 Gb of raw sequence data using four SMRT Cells of PacBio which provided ~93X fold coverage of the *V. mungo* genome (Table S1). The initial de novo assembly of long reads was performed using Canu (Koren et al, 2017) assembler at default parameters. The primary assembly consisted of 304 contigs with a total length of 479 Mb (accounting for ~85% of the estimated genome size), an N50 of 5.49 Mb, and a longest contig of 18.06 Mb. The primary contigs were further polished and error corrected by aligning SMRT reads to the assembled draft genome using pbalign and arrow. A second round of error correction was performed with 800 million pair-end Illumina short reads corresponding to ~125 Gb data and providing a genome coverage of 210X using Pilon (Walker et al, 2014). Finally, a chromosome-level genome assembly of *V. mungo* was generated based on 3D proximity information obtained via Hi-C contact reads (Fig 1A).

In total, ~158 Gb of Hi-C sequencing data (Table S2) containing 169 million contact reads were used to construct pseudochromosomes using 3D-DNA program (Dudchenko et al, 2017). Consequently, 98.7% of the assembled sequence was placed on 11 chromosomes of *V. mungo* genome (Fig 1B). The quality of assembly was assessed by using three different genome metrices: (I) We mapped Illumina DNA and RNA short reads against the assembled genome. This resulted in the alignment of 95% DNA reads and 79–97% RNA reads suggesting high nucleotide accuracy after error correction of the genome (Table S3). (II) The completeness of the assembly was evaluated by using Benchmarking Universal Single-Copy Orthologs, which revealed that 1,558 (~96.5%) out of 1,614 complete available embryophyta Benchmarking Universal Single-Copy Orthologs were represented in the assembled genome, displaying a high level of genome completeness (Table S4). (III) The long terminal repeat (LTR) assembly index (LAI) score of the genome assembly was calculated and found to be ~18 (Fig 1C). Compared with previously published genome assembly which had LAI score ~10, N50 value 43 Mb, 94% genome completeness, and 34 Mb gap (Table S5), our assembly had a much higher LAI score of 18, N50 value 44 Mb, 96.5% genome completeness with a sequence gap of only 14 kb. Together, these results suggest an improved assembly in terms of improved continuity and reduced sequencing gaps.

Genome annotation was performed using ab initio gene prediction and homology-based searches against ESTs and protein databases of *V. angularis*, *V. unguiculata*, *G. max*, *M. truncatula*, *A. thaliana*, and *O. sativa*. To obtain high-confidence sets of gene and transcriptomic evidence for annotated genes, we generated short Illumina RNA-seq and IsoSeq reads from eight different tissues of *V. mungo*. In total, 34,643 protein-coding genes spanning 118.41 Mb of the genome with an average length of 3.4 kb were predicted (Table 1).

### Comparative genomics and genome evolution in *Vigna* species

We then leveraged our new improved de novo genome assembly to understand the evolutionary history of *V. mungo* and its close relatives. For this, we analyzed synonymous substitution rate (ks) by

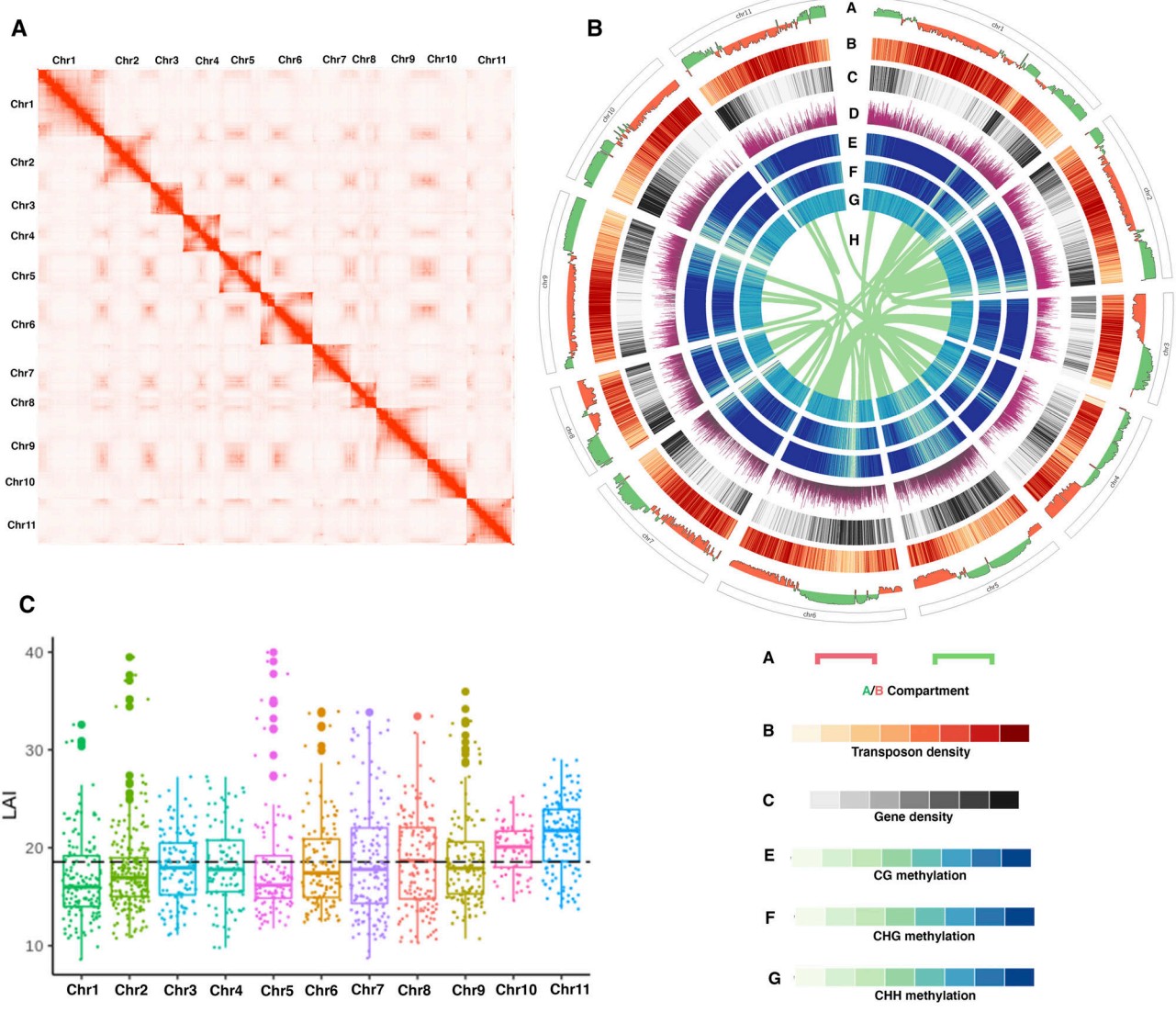

**Figure 1. An overview of the distribution of genomic features in *V. mungo* Hi-C–guided de novo genome assembly.**
**(A)** Genome-wide Hi-C contact matrix of *V. mungo*. Red color intensity in the heatmap shows frequency of interaction between two loci at 25 kb resolution. **(B)** Circos plot representing (A) A/B compartments, (B) transposon element density, (C) gene density, (D) mRNA expression, (E) CG methylation, (F) CHG methylation, (G) CHH methylation, (H) WGD genes in A/B compartment. **(C)** LAI score distribution of all 11 pseudomolecules. Dashed line indicates average LAI score 18 suggesting the high quality of assembly.

comparing duplicated genes residing in syntenic blocks within and across genomes. The density distribution of ks showed a major single peak in *V. mungo* (0.63), *V. angularis* (0.64), and *V. unguiculate* (0.67) genomes (Fig 2A), whereas the ks distribution peak of ortholog pairs ranged between 0.08 to 0.1 among these species, confirming that an ancient WGD event around 58 MYA (Yang et al, 2015) in papilionoidea preceded their speciation (Fig 2B). Phylogeny inference showed that *V. mungo* clustered with other *Vigna* species and appeared to form a monophyletic group, congruent with previous findings (Moghaddam et al, 2021). The phylogenetic analysis also revealed that *V. mungo* diverged from *V. unguiculata* 8–9 MYA, from *P. vulgaris* 11–12 MYA, and from *G. max* 25 MYA (Fig S1). Furthermore, evolution of gene families among 11 plant species indicated a substantial contraction of gene families (5,068) than

expansion (558) in *V. mungo*. We also observed that *G. max* has gained significant number of genes (+9,117/−715) compared with other species suggesting that most of the genes were retained after the WGD event. We next estimated the divergence time among *Vigna* species. For this, we used a previously reported evolutionary rate of $8.3 \times 10^{-9}$ substitutions/synonymous site/year in leguminous species (Moghaddam et al, 2021) and synonymous substitution rate (ks) among orthologs. Based on the mean peak of ks (0.08) between *V. mungo* and *V. angularis*, the divergence age was estimated to be around 4.8 MYA (Fig 2B). Using a similar approach, divergence ages between *V. mungo* and *V. unguiculata*, and *V. angularis* and *V. unguiculata* were predicted to be around 6 and 5.4 MYA, respectively. Similarly, the divergence of *G. max* from *V. mungo*, *V. angularis*, and *V. unguiculata* was estimated to be 18, 16.8, and

**Table 1.** Genome assembly and annotation statistics of *V. mungo*.

| Assembly size | 479 Mb |
|---|---|
| Chromosome number (2n) | 11 |
| No. of super scaffolds | 13 |
| Super scaffold N50 | 44 Mb |
| No. of contigs | 305 |
| Contig N50 | 8.1 Mb |
| No. of Ns (Gap) | 14 kb |
| GC content | 33.6% |
| No. of genes | 34,643 |
| Mean gene length | 3,418 bp |
| Mean CDS length | 1,164 bp |
| Mean exon length | 326 bp |
| Mean intron length | 429 bp |

17.4 MYA, respectively. These predictions are in agreement with the evolutionary relationship among legumes inferred by the phylogenetic tree (Fig S1) (Yang et al, 2015; Pootakham et al, 2021).

The inter-genomic identification of syntenic block showed a high conservation of *V. mungo* with *V. angularis*. We found that most of the chromosomes showed almost one-to-one relationship implying that chromosomes Vm02, Vm03, Vm04, Vm06, and Vm09 of *V. mungo* aligned with the chromosomes Va04, Va11, Va10, Va3, and Va02 of *V. angularis* respectively (Fig 2C). However, some chromosomes of *V. mungo*, mainly Vm01, Vm05, and Vm07 shared collinearity with more than one chromosome of *V. angularis*, indicating a structural rearrangement within these chromosomes that could be a signature of species evolution within the *Vigna* genus. Although a comparison of orthologous blocks between *V. mungo* and *G. max* indicated a high level of collinearity, however, we observed that each chromosome of *V. mungo* matched with multiple chromosomes of *G. max* (Fig S2).

In flowering plants, retrotransposon bursts are considered to be a major driver of the genome size variation during evolution (Suh, 2019). We used a combination of de novo and homology-based approaches to decipher the comparative evolutionary landscape of LTR retrotransposons in *V. mungo* and its two related species, *V. angularis* and *V. unguiculata*. Like other legumes, ~46.5% (223 Mb) of the *V. mungo* genome comprised of repetitive sequences, of which, LTRs represented the most abundant class occupying 34.6% of the genome, including 16.7% LTR/gypsy, 11.1% LTR/Copia, and 6.9% unknown LTRs (Fig 2D). We also analyzed LTR content in the abovementioned sister species using the same computational pipeline as for *V. mungo*. Consistent with its larger size, the genome of *V. unguiculata* (519 Mb) showed a higher proportion of LTRs (36.4%) followed by *V. mungo* (34.6%; LTR, 480 Mb; genome) and *V. angularis* (31.7%; LTR, 447 Mb; genome). We observed a high enrichment of Ty3/Gypsy elements (16–20%) in all the three genomes (Fig 2D). We propose that the 61% of 40 Mb genome size difference between *V. mungo* and *V. unguiculata* could be attributed to differences in the relative proportion of Gypsy elements between two species. In contrast, 90% of the 33 Mb genome size difference between *V. mungo* and *V. angularis* could be caused by an increase

in Copia (17 Mb) and unknown LTR retrotransposon (13 Mb) elements in *V. mungo* (Table S6).

To further understand the evolutionary relationship among LTRs and their potential role in genome evolution, we identified intact LTRs in all three species. In total 3,379, 3,735, and 2,732 intact LTRs were found in *V. mungo*, *V. angularis*, and *V. unguiculata*, respectively. Based on the conserved *RVE* domains of 2,580 intact Gypsy/Ty3 elements, a phylogenetic tree was reconstructed (Fig 2E). The phylogenetic analysis showed that Gypsy-RT elements amplified in clusters which could be grouped into five major clades, where each clade included elements from all three species. The phylogeny also revealed a lineage-specific differential expansion of Gypsy/Ty3 in clade 1 that might be a contributing factor to the large genome size of *V. unguiculata*. We detected a recent LTR amplification peak ranging from 0.5 to 0.8 MYA in all three species (Fig 2F). Compared with *V. mungo* and *V. unguiculata*, a recent amplification peak (0.5 MYA) was observed in *V. angularis*. This suggests an enrichment of young LTRs, which likely explains the observed high number of intact LTRs in *V. angularis*.

### Spatial 3D chromatin organization in *V. mungo*

We next aimed to resolve the 3D structure of the *V. mungo* genome. We used Hi-C interaction map to understand how the packaging of *V. mungo* genome into 3D spatial chromatin organization within the nucleus is intricately associated with the transcriptional regulation and coordinate functional activities of genes. The Hi-C interaction map showed a strong intra-chromosomal interaction frequency in adjacent loci and high inter-chromosomal frequency at the chromosomal end that confirmed the existence of chromosomal territories (CT) (Figs 1A and S3A and B). Using the first principal component (eigenvector) of the normalized contact matrix, we delineated the *V. mungo* chromatin within a specific CT at megabase resolution into two distinct A/B compartments. We found that the A compartment covered around 45% (217 Mb) of the genome and was mainly located at the arms of the chromosome, whereas the B compartment encompassed the heterochromatic region covering 55% (254 Mb) of the *V. mungo* genome (Supplemental Data 1). We also observed that chromatin regions in either A/B compartment showed a higher and similar interaction frequency than those of any two sites located between compartments or chromosomes (Figs 3A and S3A).

We next evaluated the association of two spatially separated compartments with genomic and epigenomic features (Fig 3B–G). By computing DNA methylation patterns (Table S7), we found that the B compartment showed an association with heterochromatic features such as high level of methylation in all CG, CHG, and CHH contexts and high transposon density (Fig 3D–G). In contrast, the A compartment was associated with euchromatic features and showed a significantly high number of genes and high expression (Fig 3B and C). Overall, our study demonstrates that Hi-C–generated contact matrix can be used to delineate chromosomes into euchromatin and heterochromatin regions. Our findings reconcile with previous reports of compartment characterization in plants and animals, suggesting that features of 3D chromatin compartments might be conserved across eukaryotes (Lieberman-Aiden et al, 2009; Dong et al, 2017).

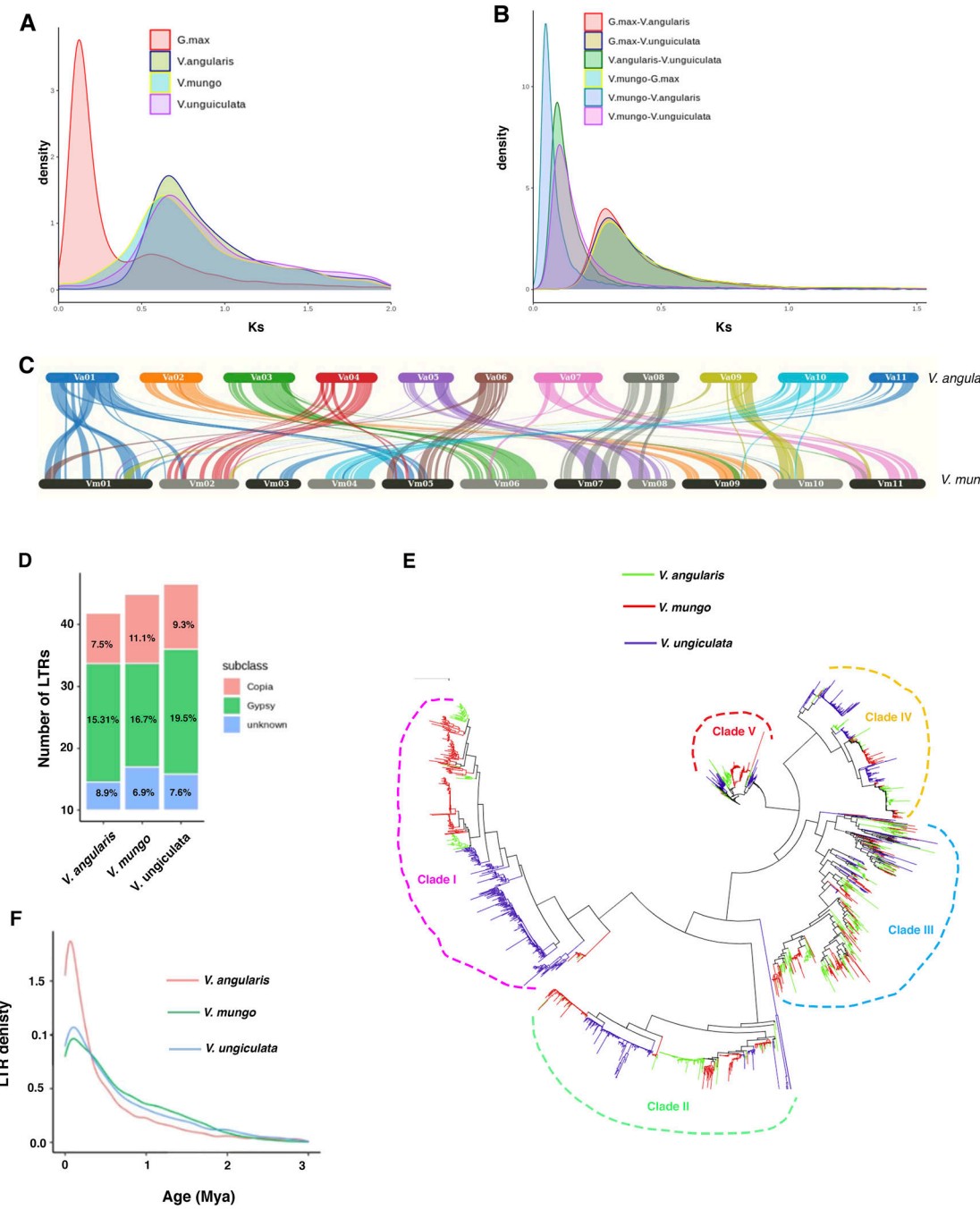

**Figure 2. A comparative analysis of genome and transposons evolution in _V. mungo_.**
**(A)** A density distribution plot of Ks values of paralogous genes showing duplication events among four species: _V. mungo, V. unguiculata, V. angularis,_ and _G. max._ **(B)** Ks density distribution plot of orthologous genes representing species divergence among species mentioned in (A). **(C)** Synteny plot showing collinear blocks and chromosomal rearrangement between _V. mungo_ and _V. angularis._ **(D)** A bar plot showing the number of LTR in _V. mungo, V. angularis,_ and _V. unguiculata._ **(E)** Phylogenetic tree of Gypsy elements. Neighbor-joining and unrooted trees were generated based on _RVE_ domains of Gypsy elements in three _Vigna_ species. **(F)** A density plot showing estimated insertion time of intact LTRs (MYA, million years ago).

We next investigated the existence of TAD-like structures (TLRs hereafter) in the _V. mungo_ genome by using HiCExplorer (Ramirez et al, 2018) and Juicer tools (Durand et al, 2016) at 10-kb resolution Hi-C contact map. HiCExplorer detected continuous TLRs with shared boundaries resulting in 1,548 TLRs with a median size of

260 kb, covering 96% of the genome (Supplemental Data 1), whereas Juicer predicted 567 discretely spaced TLRs having a median size of 220 kb, occupying 31% of the genome length (Fig 4B and C). Two different methods showed 169 overlapped TLRs covering 50 Mb of _V. mungo_ genome (Fig 4B). We found

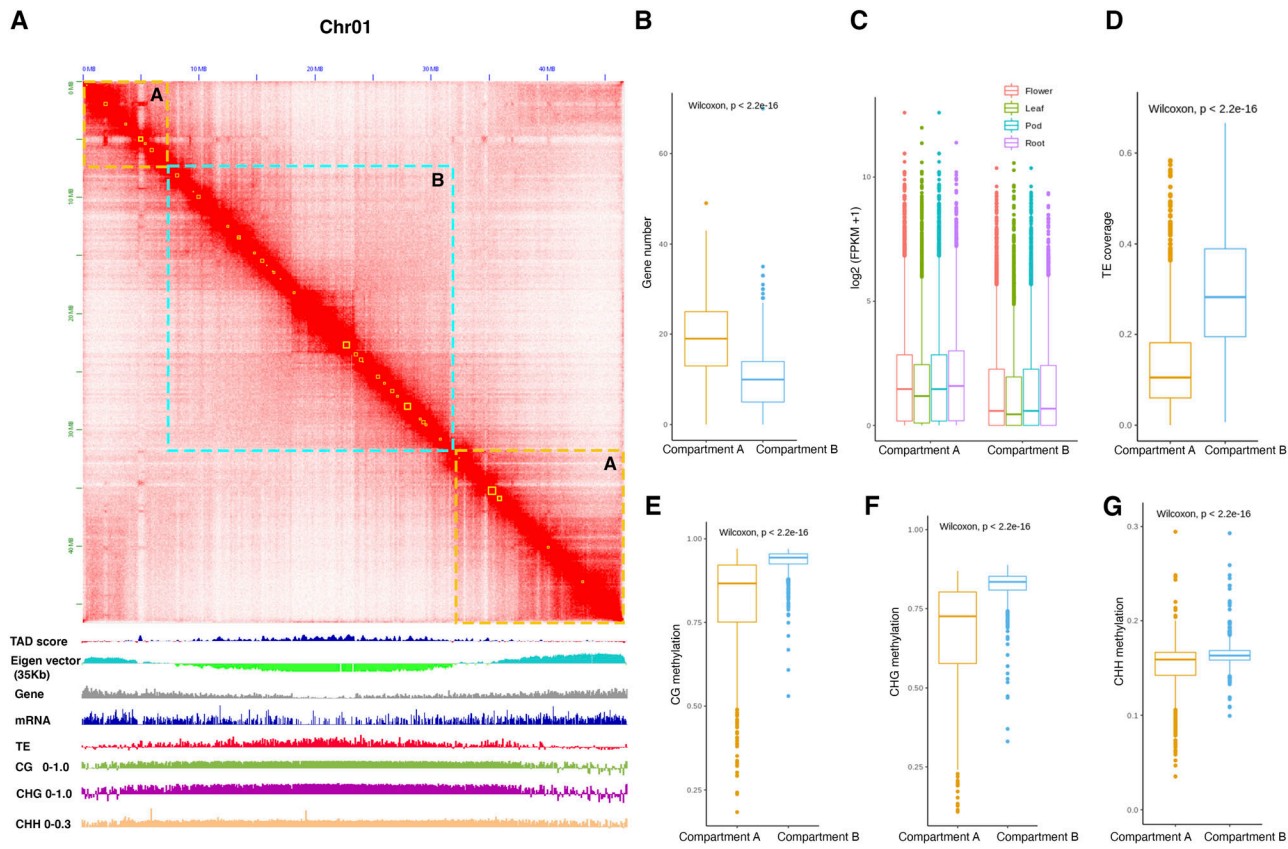

**Figure 3. Characterization of global patterns of 3D chromatin organization in *V. mungo*.**
**(A)** Hi-C contact matrix, TAD score distribution, A/B compartment based on Eigenvector (100 kb windows) and IGV genome browser showing gene and transposon density, mRNA expression, and DNA methylation marks (250-kb windows) track throughout chromosome Chr01. **(B, C, D, E, F, G)** Genome-wide distribution of genomic and epigenetic pattern in A/B compartment. **(B, C, D, E, F, G)** Euchromatic features such high gene density (B) and expression level (C) were associated with the A compartment, whereas the B compartment showed heterochromatic features such as high TE density (D) and high level of CG (E), CHG (F), and CHH methylation (G). Displayed *P*-values were calculated using Wilcoxon rank sum test with a *P*-value threshold of *P* < 0.5.

52% and 69% of TLRs annotated by HiCExplorer and Juicer, respectively, were distributed in the A compartment (Fig 4D). Visual inspection of predicted TLRs indicated a high long-range interaction frequency within the domain and a low TAD score at the boundary region of two consecutive TLRs (Fig 4A). A low TAD score indicates stronger boundaries and active TADs (Ramirez et al, 2018). Moreover, boundary-associated genes showed a significant increase in expression levels compared with genes residing within the domain (Fig 4E). We next compared the expression level of each gene across different tissues and computed the coefficient of variation (C.V). Genes located in the boundary region tend to show stable expression levels (low C.V) (Fig 4F). We also observed low transposons density at predicted TLR boundary regions (Fig 4G). Consistent with transposon density, we found low CG, CHG, and CHH methylation levels in the TLR boundary regions. Contrary to the boundary regions, the domain regions showed high DNA methylation in all three contexts (Fig 4H–J).

## Evolutionary features of TLRs among legumes

TADs have been proposed to be an "inherent property" of mammalian genomes as these domains are known to be highly

conserved across mammalian species (Dixon et al, 2012). There are evidences that within the plant kingdom, TADs may not be strictly conserved between species (Dong et al, 2017; Wang et al, 2021a). To investigate whether local chromatin architecture remained conserved within legumes, we selected *V. mungo, P. vulgaris, G. max*, and *G. soja* that have different evolutionary relationships, covering almost 15–20 million years of evolution. These species provide an excellent system to understand how the whole genome duplication events followed by either presence (*G. max*) or lack of recent diploidization (*V. mungo, P. vulgaris*) influenced the local chromatin architecture during evolution. We compared the distribution of TLRs between orthologous regions of *V. mungo, G. max, G. soja*, and *P. vulgaris*. We used available Hi-C data of *G. max, G. soja*, and *P. vulgaris* (Wang et al, 2021a) and annotated the TLRs using HiCExplorer pipeline, similar to the *V. mungo* data for consistency. In total, 3,505, 3,479, and 1,941 TLRs were identified for *G. max, G. soja*, and *P. vulgaris*, respectively (Supplemental Data 1). Next, we conducted an interspecific comparison of the genomic regions of the TLR domains and boundaries and identified 12.2–44.7% syntenic counterparts for the domain and 26–62% for boundaries. We then found that 3.5–14% of the domain and 7–20% of the boundary regions of *V. mungo* TLRs were conserved in *G. max* and *P. vulgaris*. We also

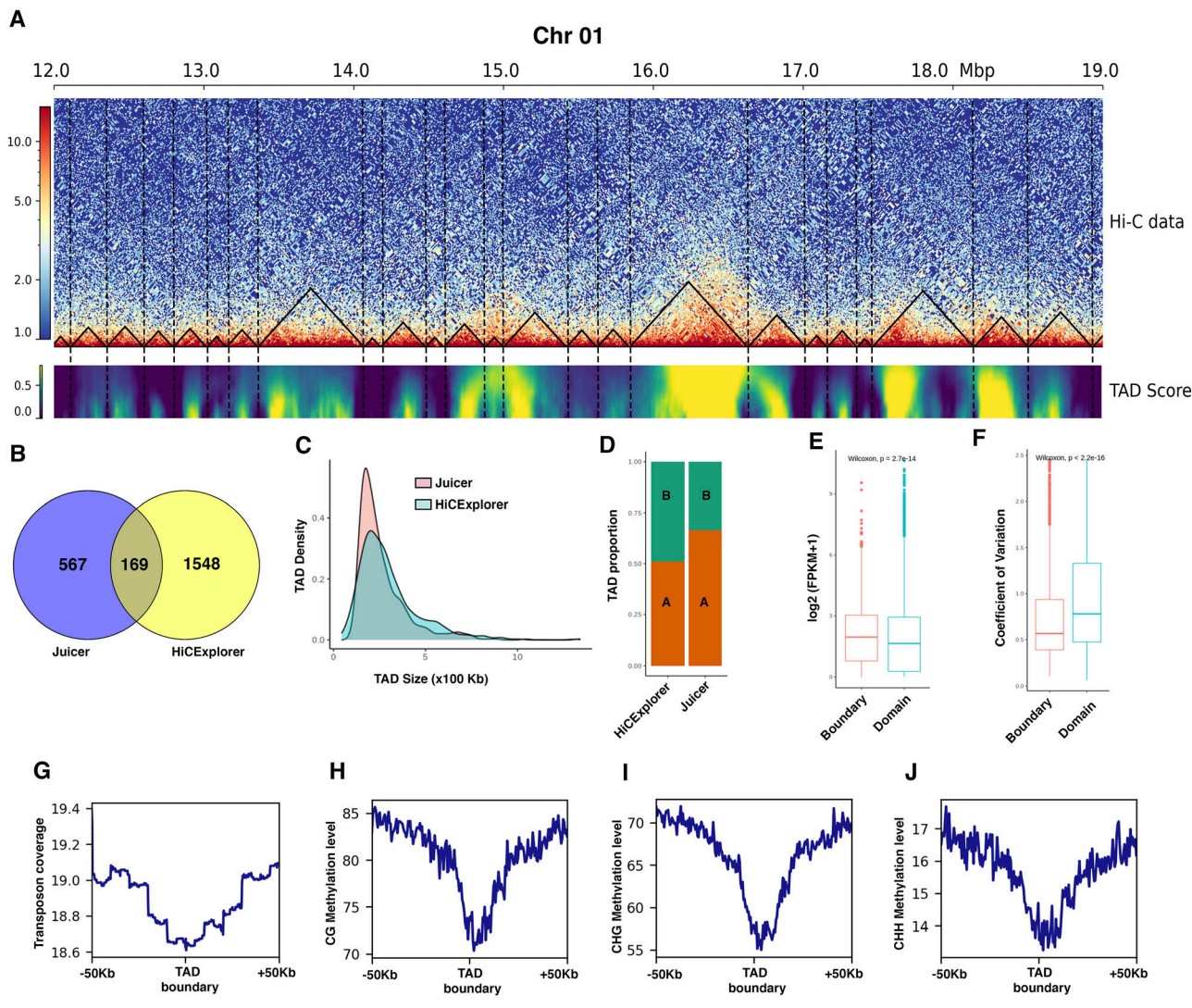

**Figure 4. TAD profiles in *V. mungo*.**
**(A)** An example region from *V. mungo* genome showing TLR structures with a resolution of 10 kb. TLRs are outlined in black solid lines. TLR boundaries are defined by the TAD separation score that ranges from 0 to 1 shown in the bottom track. **(B)** A venn diagram showing the number of TLRs and overlapped TADs identified by HiCExplorer and Juicer tool. **(C)** A density plot showing TLR size distribution for HiCExplorer and Juicer tools. **(D)** A bar plot showing the proportion of TLRs in genomic compartments. **(E, F)** Box plots showing normalized mean expression (E) and variation of expression level (F) in different tissues for genes located within TLR domains and boundary regions (50 kb upstream and downstream). **(G)** Line plot showing the distribution of transposons around TLR boundaries. **(H, I, J)** Line plots showing DNA methylation pattern in CG (H), CHG (I), and CHH (J) contexts across TLRs.

observed an almost similar trend between *G. max* and *P. vulgaris* where only 8.8% of the domain and 9.2% of the boundary regions of the TLRs were conserved (Fig 5A). Our results suggest an overall low level of conservation that was more prominent in the domains than the boundary regions of the TLRs within the legume family. We asked whether these patterns remained consistent in *G. max* and its wild relative *G. soja*. We found 55% conservation in the domain regions and 63% conservation within the boundary regions, indicating that TLRs likely remained highly conserved during domestication.

We next investigated the evolutionary features of the boundary regions and domains of the TLRs relative to the non-boundary regions within legumes. Alignment of conserved syntenic sequences from *V. unguiculata*, *G. max*, and *P. vulgaris* to the *V. mungo* genome showed relatively higher coverage in the boundary regions

compared with the non-boundary regions (Fig 5B–D). These findings indicate higher sequence conservation at the TLR boundaries than in the non-boundary regions in *Vigna* species. These patterns are similar to those observed in *solanaceae* family (Liao et al, 2022) indicating that the sequence conservation of the boundary regions may be a common feature of TLRs in plants. In metazoans, chromosomal rearrangement breaks preferentially occur at TAD boundaries and are depleted in TAD bodies. This feature of metazoan TADs constrains large-scale genome evolution (Fishman et al, 2019; Renschler et al, 2019; Liao et al, 2021). Recently, this pattern was shown to also exist in plants, in members of *solanaceae* family (Liao et al, 2022). To investigate whether this feature of TADs has also been deployed in legumes, we identified genome synteny breaks between *V. mungo*, *P. vulgaris*, and *G. max*. We found an enrichment of synteny

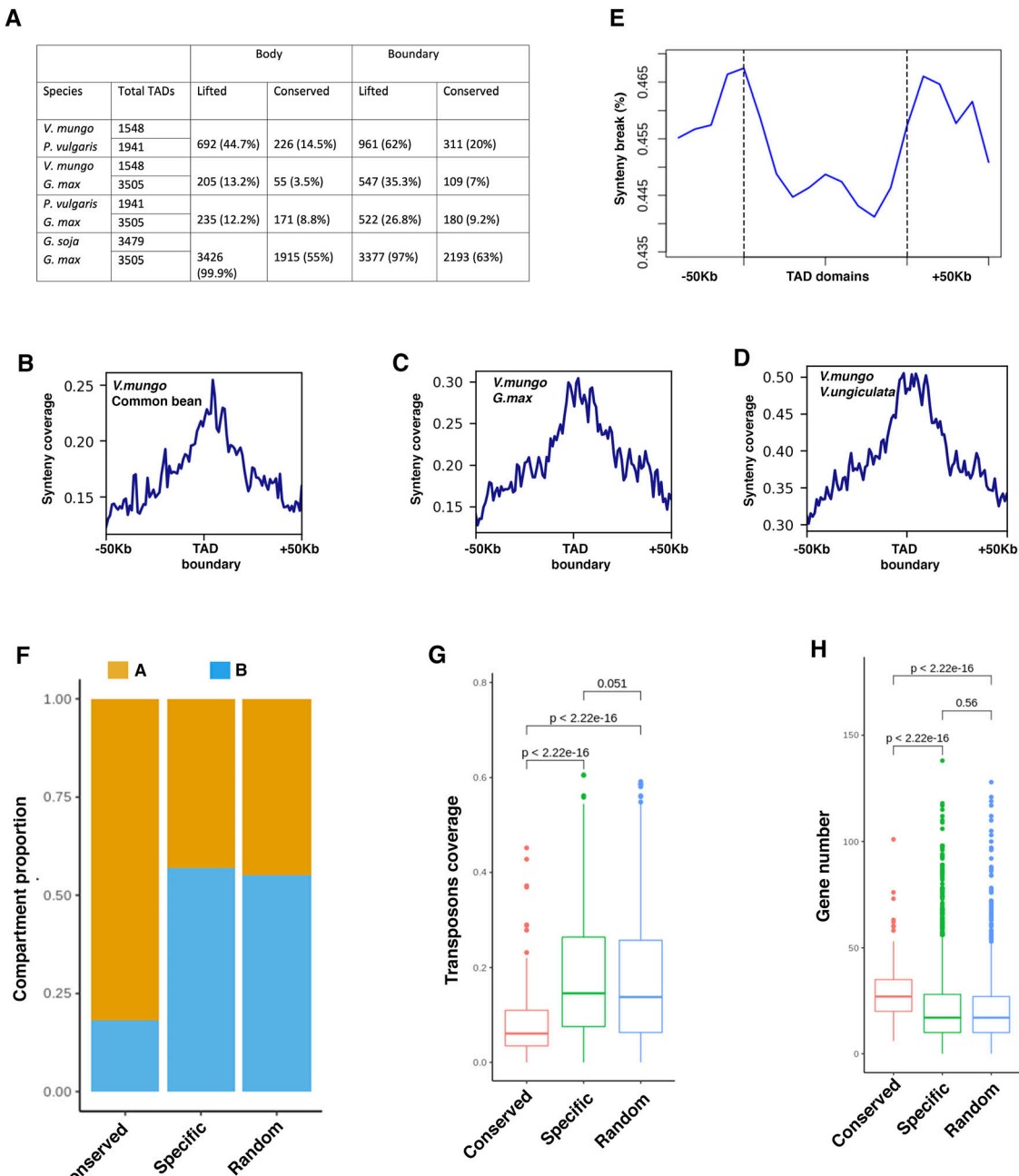

**Figure 5. Characterization of TAD conservation and evolutionary features in legumes.**
**(A)** Table showing TLR annotation and conservation summary during evolution in *V. mungo*, *P. vulgaris*, *G. max*, and *G. soja*. **(B, C, D)** Line plots showing coverage profile of conserved aligned sequence between closely related genomes of *V. mungo* and *P. vulgaris* (B), *V. mungo* and *G. max* (C), and *V. mungo* and *V. unguiculata* (D) along TLR boundaries (HiCExplorer) of *V. mungo*. **(E)** A line plot showing the distribution of syntenic breaks between *V. mungo* and *P. vulgaris* around TLRs (−50 Kb upstream and +50 Kb downstream of boundary). **(F)** Proportion of conserved and specific TLRs in chromosome compartment (A, yellow; B, blue). Randomly shuffled TLRs were used as control. *P*-values were calculated using two-sided two proportions *z*-test. **(G, H)** Box plots showing transposon coverage (G) and gene number (H) in conserved, specific, and randomly shuffled TLRs. *P*-values were calculated between each comparison by using Wilcoxon rank-sum test with a *P*-value threshold of *P* < 0.5.

breaks at the TLR boundaries identified for each comparison between *V. mungo* and two legume species, despite high evolutionary conservation of sequence at those regions in the *V. mungo* (Figs 5E and S4). Our findings indicate that enrichment of chromosomal rearrangements break at boundaries of chromatin-folding domains perhaps may be a unifying feature of TAD-like domains in animals and plants.

We then asked how the conserved genome topology relates to the specific genomic features and gene properties. We found a significant higher proportion of 82% of conserved TLRs were

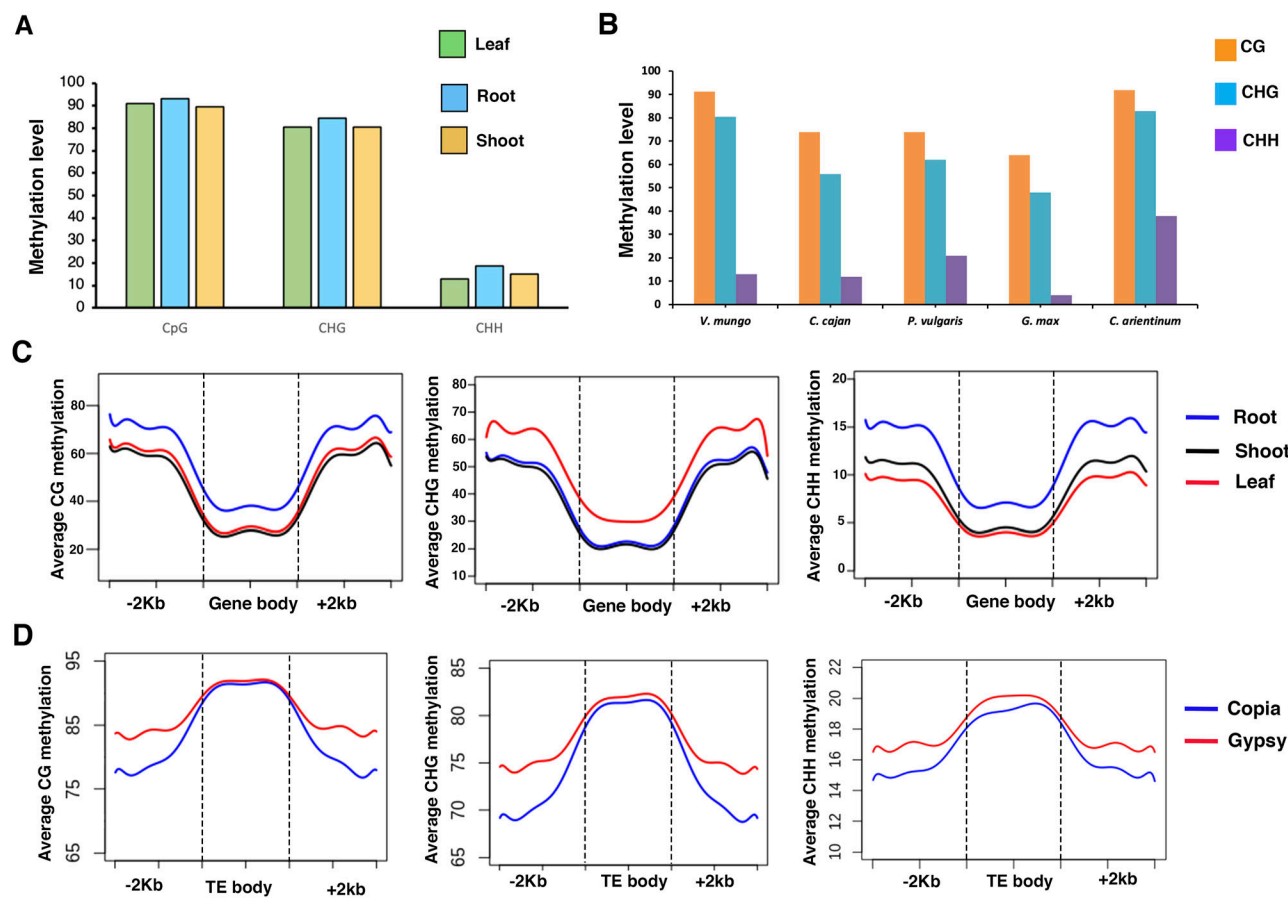

**Figure 6. Genome-wide analysis of DNA methylation in *V. mungo*.**
**(A)** Bar plots showing weightage DNA methylation level in leaf, root, and shoot. **(B)** Bar plots showing relative proportion of genome-wide methylation levels at CG, CHG, and CHH contexts in leaf tissues among different legumes. **(C, D)** Metaplots showing distribution of DNA methylation level in CG, CHG, and CHH contexts across gene (C) and transposon bodies (D) and flanking region 2 kb upstream from the transcription start site and 2 kb downstream from the transcription termination site.

present within the transcriptionally active A compartment, compared with specific TLRs (43%; two-sided two proportions *z* test *P*-value = 2.2 × 10$^{-16}$) or random regions (45%; *P*-value <2.2 × 10$^{-16}$) (Fig 5F). We also observed that species-specific TLRs showed a significantly high proportion of TEs relative to the conserved boundaries (Fig 5G). Consistently, we found that species-specific domains have low gene density (Fig 5H) and were more enriched in the B compartment than in the A compartment.

## Global role of DNA methylation on gene regulation and transposons silencing

To gain insights into the genome-wide DNA methylation landscape and its relationship with gene expression in *V. mungo*, we generated DNA methylome profiles of leaf, shoot, and root tissues in two biological replicates at single base resolution by using whole-genome bisulfite sequencing. A comparison of global weightage DNA methylation levels at symmetric (CG, CHG) and asymmetric (CHH) sites among three tissues showed highest methylation in the root (93.3% CG, 84.6% CHG, and 18.6% CHH) followed by leaf (91.2% CG, 80.6% CHG, and 13% CHH) and shoot (89.6% CG, 80.4% CHG, and 15.3% CHH) (Fig 6A).

Comparative analysis of methylation patterns within legumes indicated a much higher methylation in CG and CHG contexts within *V. mungo* leaf compared with leaves of other legumes *C. cajan* (74%, 56%) (Junaid et al, 2018, 2022), *P. vulgaris* (74%, 62%), and *G. max* (64%, 48%) (Kim et al, 2015) (Fig 6B). We observed almost similar levels of CG and CHG methylation levels as observed in *C. arientinum* (92%, 83%) (Bhatia et al, 2018). Interestingly, DNA methylation levels at asymmetric sites varied considerably, with highest in *C. arientinum* (38%) followed by *P. vulgaris* (21%), *V. mungo* (13%), and *G. max* (4%) (Fig 6B). Contrary to previous reports of a positive correlation between the genome size, transposons contents, and DNA methylation, no such correlation within legumes were evident in our analysis. Large-sized genomes tend to be highly methylated as a consequence of methylation-enriched TE abundance (Alonso et al, 2015; Niederhuth et al, 2016). Despite an obvious size variation in the genomes of abovementioned species, we did not find a substantial difference in their transposon content (Fig S1), which possibly explains the observed methylation patterns (Fig 6B). We also examined DNA methylation patterns in the gene body and flanking regions (2 kb upstream from TSS and 2 kb downstream from TTS) in *V. mungo* (Fig 6C). Average methylation levels were significantly high in the flanking regions than in the gene body regions in

all three cytosine sequence contexts. Consistent with whole-genome methylation levels, we observed highest CG and CHH methylation in root followed by leaf and shoot. Average CHG methylation was highest in leaf, whereas root and shoot showed almost similar levels of methylation in the flanking and gene body regions.

Having characterized DNA methylation patterns extensively, we next investigated the role of gene body methylation and its relationship with gene expression. We first categorized non-TE genes based on their methylation levels. The normalized expression level of each gene category was assessed using RNA-seq data from the leaf samples (age, 45 d after germination). Genes with low (10–20%) and intermediate levels (20–40%) of CG body methylation showed high expression, whereas moderately (50–70%) and heavily methylated genes (>70%) showed low expression (Fig S5). This suggests that low and intermediate levels of methylation might enhance transcription by increasing splicing efficiency of the primary transcripts. In contrast, we observed a negative association of CHG and CHH methylation levels with gene expression.

Integration of transposons substantially contributes to the genome size variation and also has a potential to affect gene expression (Kidwell, 2002; Lee & Kim, 2014). Cytosine methylation represses transposons activity, rendering them immobile, which can be co-opted by the host genome and potentially contributes to an evolutionary innovation over a time period (Zhou et al, 2020). To understand the evolutionary impact of LTRs on DNA methylation, we examined methylation patterns in the transposon body and flanking regions (Fig 6D). Metaplot analysis showed highest methylation level in the transposon body regions in all cytosine contexts. Gypsy elements displayed slightly higher methylation in the body and flanking regions compared with copia elements. We also observed a depletion of methylation level in the flanking region relative to the transposon body in both elements. Furthermore, we investigated the relationship between LTRs age and methylation levels and found a significant negative correlation in all the three cytosine sequence contexts (Fig S6A–C). These findings indicate high methylation levels in young LTRs in *V. mungo* genome, a pattern also observed in other plant species (Choi & Purugganan, 2018; Wang & Baulcombe, 2020).

## Discussion

In this study, we generated a high-quality, nearly complete chromosome-scale reference genome assembly of 479 Mb (85% of estimated genome size) of *V. mungo* and identified 34,643 high-confidence protein-coding genes. Compared with two previous published assemblies, our assembly showed improved assembly continuity (high LAI score of 18) and completeness. In addition, we also developed a 3D chromatin map of *V. mungo* genome by integrating genomic, transcriptomic, and epigenomic data from different tissue types. Our findings shed light into epigenetic patterns and spatial 3D genome organization of *V. mungo* that had not been studied until now.

Understanding complex genome organization and chromatin topology in higher eukaryotes is an important but a nontrivial problem. Here, we used a Hi-C–generated contact matrix to perform

sub-compartment annotation of *V. mungo* chromosomes to resolve euchromatin and heterochromatin regions. We found that like mammalian and other angiosperm genomes (Lieberman-Aiden et al, 2009; Dong et al, 2017), *V. mungo* genome could be partitioned into global A/B compartments. This compartmentalization of *V. mungo* genome helped resolve distinct genomic and epigenomic features that are associated with these compartments. First, the A compartment is mainly associated with euchromatin regions and is located at chromosomal arms, whereas the B compartment mainly contains heterochromatin and is located at the centromeric and pericentromeric regions of chromosomes. Second, the B compartment correlates well with enrichment in DNA methylation and TE density, whereas the A compartment shows signatures of gene enrichment and low DNA methylation. We conclude that in *V. mungo*, compartments A and B associate with transcriptionally active and dormant states of chromatin, respectively. Our results further indicate that high abundance of TE in B compartment may be a possible mechanism of keeping the chromatin in a dormant state. This regulatory mechanism deviates from that reported in the pepper genome recently where contrasting patterns of methylation were reported, that is, high methylation in transcriptionally active A compartment than in the transcriptionally quiescent B compartment indicating that in this case, transcriptional dormancy of B compartment may be because of perhaps a higher proportion of fully silenced TE compartments that are less likely targets of methylation (Liao et al, 2022).

At sub-megabase scale, mammalian genomes are folded into TADs via a cohesion–CTCF loop extrusion mechanism, which exhibits a high intra-TAD interaction frequency (Dixon et al, 2012; Sexton et al, 2012). Plant genomes have been found to display TAD-like features; however, their establishment is proposed to be conducted via a non-cohesion–CTCF loop extrusion mechanism (Dong et al, 2017; Rowley et al, 2017). Here, we report the presence of TAD-like regions in the *V. mungo* genome that cover almost 96% of the full genome. We further deconvolute genomic and epigenomic features of the TLRs and their boundary regions. Specifically, we show that boundaries of TLRs in *V. mungo* are a hotspot for transcriptional activity as they house highly expressed genes corresponding with low methylation levels, whereas TLRs mainly correspond with transcriptionally inactive chromatin that is enriched in DNA methylation. This interspersed organization of transcriptionally active and inactive domains has also been observed in other crops such as pepper (Liao et al, 2022) and wheat (Concia et al, 2020), suggesting that this may likely be a common mechanism for spatially clustering transcriptionally active genes within plants.

We found low conservation of TLRs within *Vigna* species, a pattern aligning with that observed in multiple plant species including *Arabidopsis*, foxtail millet, sorghum, rice, and maize, and between close relatives—soybean and common bean (Dong et al, 2017; Wang et al, 2021a) and contrary to animals (Dixon et al, 2012; Vietri Rudan et al, 2015). Our study shows that most TLRs coincide with heterochromatin regions that are also enriched in rapidly changing transposing elements, the dynamics TE content may influence TLR conservation. By zooming in further within the structure of TLRs, we found relatively high conservation in the boundaries of TLRs than in the domains (Fig 5A). Notably, TLR boundary regions

correspond with low TE coverage than the domain regions. Our findings therefore help support the observed inverse relationship between TLR conservation and TE content (Fig 5G). However, our results along with previous work in pepper (Liao et al, 2022) and *Drosophila* (Liao et al, 2021) also present a paradox that the genome arrangement breakpoints or synteny breaks also occurred in the TLR boundary regions that are under sequence constraints (Fig 5B–D). It has been proposed that higher "chromatin fragility" at domain boundaries may give rise to chromosomal breaks at TAD borders than at the genome background (Berthelot et al, 2015). Therefore, future studies targeted towards identifying and characterizing genome-wide synteny breakpoints will be important to help resolve this paradox.

# Materials and Methods

### Plant material and sequencing

The cultivar *V. mungo* var. IPU0243 was used in this study for de novo assembly and genome annotation. Plants were grown in the greenhouse facility at NIPGR, New Delhi, under long day conditions (16 h light and 8 h dark). DNA was isolated from fresh young leaves of 20-d-old plants using the QIAGEN DNeasy Plant Mini Kit following the manufacturer's protocol (QIAGEN). High-quality DNA was used for library preparation according to the manufacturer's protocol (Illumina) and library was sequences on HiSeq X platform in pair-end mode. For PacBio sequencing, high molecular weight DNA extracted from young fresh leaves for construction of PacBio SMRT library following the standard SMRT bell construction protocol (Pendleton et al, 2015), and the library was sequenced on a PacBio Sequel platform. Hi-C library preparation was carried out following the protocols implemented by Dovetail Genomics and subsequently sequenced on an Illumina HiSeq X platform. For transcriptome sequencing, high-quality RNA was isolated in two biological replicates from leaf (24 d and 45 d), root (24 d and 45 d), flower, young pod tissues using QIAGEN RNeasy Kit following the manufacturer's protocol. Quality and quantity assessment of isolated RNA was done through Agilent Bioanalyzer 2100 (Agilent Technologies) and qubit *V. mungo* was used for strand-specific RNA-seq library preparation using TruSeq sample preparation kit. RNA-seq libraries were subjected to sequencing on the Illumina HiSeq X platform to generate 150-bp paired-end reads. Reads were then processed for quality filtering with TrimGalore (https://github.com/FelixKrueger/TrimGalore) to remove adapters and low-quality sequences, and the processed filtered reads were aligned to the assembled *V. mungo* genome using HISAT2 (v2.1).

The published genomic data of *G. max* (SRX8986037, SRX8986036), *P. vulgaris* (SRX8986009, SRX8986008), and *G. Soja* (SRX8986019, SRX8986018) were used in this study and subjected to similar analysis as *V. mungo*.

### Genome assembly and compartment analysis

The long raw reads were assembled by Canu assembler at the default parameter with the following phases: (a) correction of

bases, (b) trimming of low quality and adapter sequences, (c) assembly. The initial draft assembly was polished and error corrected in two steps; first, PacBio reads were aligned to the reference assembly using pbalign and the output aligned file was used to run Arrow (https://hprc.tamu.edu/wiki/Bioinformatics:PacBio_tools) for polishing of draft assembly. Second, the short Illumina reads were mapped to the polished assembly using BWA (Li & Durbin, 2009) and error correction was performed using Pilon. We next used Hi-C reads to scaffold PacBio-generated assembly into pseudomolecules by using 3D-DNA pipeline. The raw Hi-C reads were processed for adapter and low-quality sequence removal using TrimGalore. 3D-DNA pipeline was run in iterative mode to remove misjoins, order and orient the input scaffolds. To generate Hi-C contact map, cleaned read pairs were aligned back to the assembly and the Hi-C contact map was build based on valid interaction pairs using Juicer program (Durand et al, 2016). The Hi-C contact reads were used to study spatial organization of chromatin in 3D genome space. For this, linear genome was partitioned into fixed size loci at 10 kb resolution and interaction matrix was created. The raw contact matrix was normalized to resolve the biases that arise because of frequent chromatin interaction between loci and PCR amplification as described previously (Lieberman-Aiden et al, 2009). The normalized contact matrix was converted into a binary hic format for visualization and downstream analysis in Juicebox (Durand et al, 2016). Then, eigenvector implemented in Juicer tools was used to identify A/B compartments in the genome at 100 kb resolution. The genome-wide contact domains or TADs were annotated using Arrowhead algorithm with KR (Knight & Ruiz, 2013) normalization method at 5 and 10 kb resolution.

For identification of TAD feature (i.e., body and boundary) conservation, the generated syntenic chain file, and UCSC liftOver command were used for conversion of genomic coordinates between species. For a feature to be successfully lifted over, it should have 25% minimum ratio of bases (−minMatch = 0.25) for body and 0.33 minimum match for boundary remapped to other species. BEDTools intersect tool (Quinlan & Hall, 2010) with parameters: −f 0.8 −r was used for conserved TAD body identification. Furthermore, for boundaries, any overlap was considered as indicative of conservation.

### Transcriptome assembly and genome annotation

We used TrimGalore to remove adapter sequences and filter out low-quality reads from RNA-seq data. For Iso-seq long reads processing, we applied the IsoSeq v3 pipeline (https://github.com/PacificBiosciences/IsoSeq). Briefly, Circular Consensus Sequences were generated from each individual subreads. The demultiplexing and removal of primer sequences from Circular Consensus Sequence reads were done using *lima*. Next, PolyA tails were trimmed and clustering steps were performed to generate polished isoforms.

For genome annotation, we applied two approaches to identify high confidence genes: (1) RNA-seq paired-end reads and Iso-seq long reads were combined to run trinity pipeline (Haas et al, 2013) for de novo transcripts assembly and (2) RNA-seq reads were aligned to reference genome and transcripts were reconstructed using cufflink pipelines (Trapnell et al, 2012). These two sets of transcripts were combined and redundant transcripts with minimum identity of 0.95% were clustered or removed using cd-hit

program (Fu et al, 2012). The full proteome and EST sequences from reference species were downloaded from NCBI. Then MAKER (Cantarel et al, 2008) was run in two rounds for ab initio gene prediction. In the first round, repetitive elements were masked and proteome and transcript sequences were aligned to the reference genome using BLAST (Altschul et al, 1997) and exonerated (Slater & Birney, 2005) for initial gene prediction. In the second round, SNAP, Augustus (Stanke et al, 2006), and GeneMark tools (Besemer et al, 2001) were used to train gene models.

### Gene family classification and phylogenomic analysis

To analyze gene family and orthogroups, protein-coding genes from 11 plant species including *A. thaliana*, *O. sativa*, *M. truncatula*, *C. cajan*, *G. max*, *V. unguiculata*, *V. radiata*, *V. angularis*, *C. arientinum*, *A. duranensis*, and *P. vulgaris* were used to run Orthofinder (Emms & Kelly, 2019) at default parameters. The orthogroups containing single-copy genes were identified and a rooted phylogenetic tree was constructed and visualized using IQ-TREE (Nguyen et al, 2015) and iTOL (Letunic & Bork, 2021) programs, respectively. The phylogenetic tree in Newick format was exported to MEGA (Tamura et al, 2021) to estimate divergence time for each node. A calibration time was calculated between *M. truncatula* and *C. arientinum* using TimeTree (Kumar et al, 2017). Then, calibration constrain was added to launch MEGA run using General Time Reversible model. We next analyzed rate of gain or loss of gene families using a probabilistic graphical model in the above-mentioned species. For this, a matrix containing gene family size and Newick formatted phylogenetic tree was used as input to run CAFÉ5 program (Mendes et al, 2021).

### Whole genome duplication and syntenic analysis

For whole genome duplication analysis, pairwise homologs were identified between *V. mungo*, *V. angularis*, *V. unguiculata*, and *G. max* using reciprocal BLASTP approach. The protein sequences of duplicated genes were aligned using MAFT program and synonymous substitution rates (ks) were determined using the Nei & Gojobori model (Nei & Gojobori, 1986) in Ka-Ks calculator (Zhang et al, 2006). The ks distribution plots between *V. mungo* and other species were plotted using ggplot2 package in R environment. For syntenic analysis, we used MCScanX (Wang et al, 2012) to identify collinear blocks at default parameters. The synteny plots of the collinear region were generated by using SynVisio (https://synvisio.github.io/#/).

### Annotation of repetitive elements

To identify repeat elements, a de novo library of repetitive sequences in the genome was constructed using Repeatmodeler. Then, repeat library was used to screen and annotate repetitive sequences using RepeatMasker at default parameter. We used LTR_retriver (Ou & Jiang, 2018) that takes LTRharvest (Ellinghaus et al, 2008) and LTR_FINDER (Xu & Wang, 2007) screened LTRs as input to identify intact LTR retrotransposons in the genome. The insertion age of intact LTRs was estimated based on divergence time (K) and mutation rate (μ). The divergence rate (K) in noncoding sequences was determined by applying Jukes–Cantor model and mutation rate was assumed to be equal to $1.64 \times 10^{-8}$ substitution per site per year. To find the RT domain, Gypsy elements from *V. mungo*, *V. angularis*, and *V. unguiculata* were scanned against hmm database using hmmscan search. The RT domains of three species were aligned against each other using MAFT program (Katoh & Standley, 2013). The phylogenetic tree was built using JTT matrix model implemented in IQ-TREE (Nguyen et al, 2015).

### BS-seq library preparation and methylation analysis

High-quality DNA was isolated from plant tissues using DNeasy Plant Mini Kit (QIAGEN) according to the manufacturer's protocol. For library preparation, genomic DNA was spiked with λ DNA and then fragmented using sonication to a mean size of ~250 bp. Fragmented DNA was A-tailed and adapter ligation was done according to the manufacturer's instructions. Ligated DNA fragments were then bisulfite converted using EpiTech Bisulfite Kit (QIAGEN). Finally, bisulfite-converted DNA fragments were PCR-amplified and sequenced at HiSeqX to generate 150-bp paired-end reads from each library. The cleaned MethylC-seq reads were aligned to the reference genome using Bismark. Only uniquely aligned reads were used to call genome-wide cytosine methylation at single-base resolution. The methylation levels in CG, CHG, and CHH contexts were defined in terms of weightage methylation levels (Schultz et al, 2012).

### Statistical analysis

All statistical tests were performed in R. Statistical analysis performed are specified in the figure legends.

# Data Availability

Data generated and used in this study has been deposited to NCBI submission SUB13898971 and BioProject PRJNA1028123. The published genomic data used and analyzed this study are of *G. max* (SRX8986037, SRX8986036), *P. vulgaris* (SRX8986009, SRX8986008), and *G. Soja* (SRX8986019, SRX8986018).

# Supplementary Information

# Acknowledgements

The research grant (BT/Ag/Network/Pulses-I/2017-18) for carrying out the work reported here was received from the Department of Biotechnology (DBT), Government of India, and is greatly acknowledged. The seeds of *V. mungo* var. IPU0243 were a kind gift from the Indian Institute of Pulses Research (ICAR-IIPR), India.

## Author Contributions

A Junaid: conceptualization, resources, data curation, software, formal analysis, validation, investigation, visualization, methodology, and writing—original draft, review, and editing.
B Singh: data curation, formal analysis, visualization, methodology, and writing—review and editing.
S Bhatia: conceptualization, supervision, funding acquisition, investigation, project administration, and writing—review and editing.

## Conflict of Interest Statement

The authors declare that they have no conflict of interest.

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
