## [Reviewer comments · Life Science Alliance]

Life Science Alliance

Evolutionary insights into 3D genome organization and epigenetic landscape of *V. mungo*

Alim Junaid, Baljinder Singh and Sabhyata Bhatia

DOI: <https://doi.org/10.26508/lsa.202302074>

Corresponding author(s): Sabhyata Bhatia (National Institute of Plant Genome Research)

Review Timeline:

Submission Date:	2023-04-04
Editorial Decision:	2023-05-25
Revision Received:	2023-08-10
Editorial Decision:	2023-09-26
Revision Received:	2023-10-17
Accepted:	2023-10-19

Transaction Report:

May 25, 2023

Re: Life Science Alliance manuscript #LSA-2023-02074-T

Sabhyata Bhatia
National Institute of Plant Genome Research

Dear Dr. Bhatia,

Thank you for submitting your manuscript entitled "A chromosome-level assembly provides evolutionary insights into 3D genome organization and epigenetic landscape of *V. mungo*" to Life Science Alliance. The manuscript was assessed by expert reviewers, whose comments are appended to this letter. We invite you to submit a revised manuscript addressing the Reviewer comments.

Thank you for this interesting contribution to Life Science Alliance. We are looking forward to receiving your revised manuscript.

Sincerely,

B. MANUSCRIPT ORGANIZATION AND FORMATTING:

Reviewer #1 (Comments to the Authors (Required)):

This manuscript describes the near-complete genome sequencing of *Vigna mungo* and also provides insights into 3D chromatin organization by interrogating Hi-C reads. In addition, transcriptome analysis, along with DMR analysis, attempted to construct genome-wide epigenomic transcriptional networks in *V. mungo*.

Although the completion of genome seq. of *V. mungo* certainly will be informative to relevant fields, the majority of analysis are largely descriptive and needs further verification to justify the conclusion.

Below are several points for the authors to consider.

1. TLR callings by HiCExplorer and Juicer tools have very few overlaps (Fig. 4D). The authors should look into more detail about what causes this discrepancy. It appears that the authors used HiCExplorer for the rest of the analysis but did not provide convincing reasoning to support it. After all, both HiCExplorer and Juicer tool were developed mainly for mammalian/ animal genomes, which may not be suitable for plant genomes.

2. Although having multiple replicates of Hi-C is not common, it may be necessary in the case of calling TLRs from relatively less characterized genomes. This would allow the authors to test whether TLR callings using either HiCExplorer and/or Juicer tool (or any other analysis tools) are reproducible.

One of the main conclusions of this manuscript is the claim that TADs are not well-conserved among related species in plants. Line: 319 - Dong et al. only compared Arabidopsis and rice; The authors should not overstate it. Nonetheless, this is an intriguing conclusion to draw but perhaps too preliminary to conclude based on TAD-callings that have not been verified properly (see point #1).

Minor:

Line 64: Unlike animals, CTCF and SMC are not found in plants, and therefore the mechanism of TADs formation in plant genome still remains debatable.

SMCs are present in plants (i.e. Lam et al., 2005. J. Cell Sci)

Reviewer #2 (Comments to the Authors (Required)):

In this manuscript, Junaid et al., investigate the evolution of 3D chromatin architecture in legumes. Authors generated a chromosome-scale reference genome assembly of *V. mungo* and then used BS-seq and a HiC-based contact matrix, to compare the epigenetic landscape and 3D chromatin architecture in legumes. As expected, authors observed both euchromatic, transcriptionally-active A compartment and heterochromatic, transcriptionally-dormant B compartment. Authors observed also the presence of TAD-like-regions (TLR) throughout the diagonal of the HiC matrix. They reported a low conservation of TLRs during evolution of legumes. The work is mainly descriptive and some conclusions are not supported by the data but will impact the legume community and, likely if the epigenetic research is significantly improved, the plant field.

1. The quality of the HiC could be improved. The cis : trans ratio is commonly used as an indicator of Hi-C library quality given that inter-chromosomal interactions are a mixture of true chromatin interactions and interactions that are the result of random ligations (<https://doi.org/10.1038/s41592-021-01248-7>)

2. Figure 4A the TAD score and the triangle presented does not fit with each other?

3. Juicer and Hicexplorer, which are two different methods for analysis of the data, showed very different results and small overlap why?

4. To investigate whether local chromatin architecture remained conserved within legumes, authors selected *V. mungo*, *P. vulgaris*, *G. max* and *G. soja*. How can we compare data of different quality and resolution? This needs to be strongly justified as now is a major weakness of the comparative genomics approach. Furthermore, it is not clear at all that HiCExplorer is the best tool for that. See: <https://doi.org/10.1186/s13059-018-1596-9>

5. Authors wrote that " To gain insight into the genome wide DNA methylation landscape as well as its relationship with gene expression in *V. mungo*, we generated DNA methylome profiles of leaf, shoot and root tissues in two biological replicates at single base resolution by using whole-genome bisulfite sequencing (WGBS)". Expected correlations are described, but the

novelty of the data in this section should be justified (just anticorrelation between DNA methylation and expression is not sufficient....)

6. Authors proposed that "low and intermediate levels of methylation might enhance transcription by increasing splicing efficiency of the primary transcripts." This is only one possibility over many others and more thorough analysis of splicing must be done (e.g. nascent RNAs to detect primary transcripts in relation with DNA methylation? Other marks?).

7. What about histone modifications?

Reviewer #3 (Comments to the Authors (Required)):

This manuscript reports assembly of a high quality, near-complete, chromosome-level reference genome of *Vigna mungo*, and its spatial 3D genome organization (Hi-C) and global DNA methylation pattern. In addition, based on these data and those from the public repository, authors conducted comparative analyses from an evolutionary perspective in legume. Results of this study provide new genomics landscapes and 3D structures of *V. mungo*, which are useful in further studies of legume plants and crops. This study was well designed and executed, the manuscript overall was well written. I am not fully confident with some detailed analyzing tools in the Hi-C part due to unfamiliarity, nevertheless, I assume these are reliable given similar patterns (but see following comments) were reported in other plants, especially legumes, such as soybean. Apart from this less certain aspect, all conclusions are fairly supported by data. I have some additional concerns and suggestions, especially concerning Figures and Figure legends.

1. Although the writing is generally good and comprehensible, there are numerous grammar mistakes and inappropriate use of terminologies, for example "transposons elements". Careful English-editing and proof-reading must be conducted before acceptance of the manuscript.

2. It was shown in *Glycine max* (Wang et al. 2021a) that for most chromosomes the A compartments tended to distribute at terminal regions, while B compartments trended towards the centromere. Author discussed this point in lines 456-460, but did not explain why *V. mungo* is so different as shown in Figure 1B.

3. The color key of heatmap in Figure 1B is missing.

4. Line 122: I cannot find (H) in Figure 1B. Does it refer to the green line? Please clarify.

5. Given comparative genomics constitutes a major content of the paper, the evolutionary relationships among *V. mungo*, *V. angularis*, *V. unguiculata* *G. max* are better to be briefly described in the Introduction, as not all readers are familiar with the taxonomy of legumes.

6. Sources of the genome data of *V. angularis*, *V. unguiculata* and *G. max* should be given in M&M.

7. Lines 170: 58 MYA is based on the current study or cited from a prior reference? Please clarify.

8. Please clarify the *V. mungo* and *V. angularis* in Figure 2C.

9. Lines 204-206: These results seem different from that of Pootakham et al. 2021. Why only focused on *V. mungo* and *V. angularis*?

10. Lines 218-219: The data and figure do not match. Something went wrong in your description and/or Figure 2D?

11. Typo in Figure 2E. Please correct "Clade V".

12. Line 238: Maybe change Figure 2F to a line chart or density plot. It is difficult to get the important information about "0.5 to 0.8 MYA" from the current Figure 2F.

13. Line 423: What does 45D mean? 45 days of leaf. There is only one replicate of 45D leaf in Table S3?

14. Lines 455-456, 471-472, and 480-482: References are needed.

15. Please uniform "Mb" or "mb", "Gb" or "GB", "HiC" or "Hi-C".

Manuscript ID: LSA-2023-02074-T

Point-by-point response to reviewer comments

Reviewer 1

1. TLR callings by HiCEXplorer and Juicer tools have very few overlaps (Fig. 4D). The authors should look into more detail about what causes this discrepancy. It appears that the authors used HiCEXplorer for the rest of the analysis but did not provide convincing reasoning to support it. After all, both HiCEXplorer and Juicer tools were developed mainly for mammalian/ animal genomes, which may not be suitable for plant genomes.

Although HiCEXplorer and Juicer tools have been developed for animal genomes, but several model and non-model plants studies related to 3D genome organization have been published by using these tools. TAD annotation may vary moderately among different computational methods (Forcato et al. 2017). These tools can yield results with different properties. For example, Arrowhead allows nested TADs and permits them to be spaced discretely, resulting in adjacent TADs separated by gaps. On the other hand, most adjacent TADs among HiCEXplorer calls share boundaries. We reckon this may be one of the reasons why the calls from different methods may not overlap completely.

2. Although having multiple replicates of Hi-C is not common, it may be necessary in the case of calling TLRs from relatively less characterized genomes. This would allow the authors to test whether TLR callings using either HiCEXplorer and/or Juicer tool (or any other analysis tools) are reproducible.

We thank the reviewer for this comment. We would like to mention that although replicates may increase the reliability of the data, it may not be the case for TLRs. For example- Forcato et al reported that “chromatin interactions identified in one replicate were found to be poorly conserved in another replicate of the same tissue” (Forcato et al, *Nature Methods*, <https://www.nature.com/articles/nmeth.4325>). It is hypothesized that biological replicates may not be identical in terms of chromatin contact when quantifying reproducibility in terms of the co-occurrence of the same point interaction. This could be due to the fact that at a given time, biological cells also may be in different states and stages of the cell cycle, as hypothesized (Forcato et al, 2017 *Nature methods*).

3. One of the main conclusions of this manuscript is the claim that TADs are not well-conserved among related species in plants. Line: 319 - Dong et al. only compared Arabidopsis and rice; The authors should not overstate it. Nonetheless, this is an intriguing conclusion to draw but perhaps too preliminary to conclude based on TAD-calls that have not been verified properly (see point #1).

Dong et al, 2017 (<https://www.sciencedirect.com/science/article/pii/S1674205217303398>) have performed TAD conservation analysis in multiple plant species including Sorghum, Foxtail millet and maize. They did not observe syntenic TAD domains in these species. Additionally, it has been reported that only 13% TADS are conserved between soybean and common bean (Wang et al, 2021, Plant Physiology). Nevertheless, we agree that these reports along with our study do not cover the pan-plant kingdom and therefore, we have modified the text to tone down on this conclusion. Lines 499-501

Minor:

Line 64: Unlike animals, CTCF and SMC are not found in plants, and therefore the mechanism of TADs formation in plant genome remains debatable. SMCs are present in plants (i.e., Lam et al., 2005. J. Cell Sci).

We thank the reviewer for pointing this out. We have removed this statement from the text.

Reviewer 2

In this manuscript, Junaid et al., investigate the evolution of 3D chromatin architecture in legumes. Authors generated a chromosome-scale reference genome assembly of *V. mungo* and then used BS-seq and a HiC-based contact matrix, to compare the epigenetic landscape and 3D chromatin architecture in legumes. As expected, authors observed both euchromatic, transcriptionally-active A compartment and heterochromatic, transcriptionally-dormant B compartment. Authors observed also the presence of TAD-like-regions (TLR) throughout the diagonal of the HiC matrix. They reported a low conservation of TLRs during evolution of legumes. The work is mainly descriptive and some conclusions are not supported by the data but will impact the legume community and, likely if the epigenetic research is significantly improved, the plant field.

1. The quality of the HiC could be improved. The cis: trans ratio is commonly used as an indicator of Hi-C library quality given that inter-chromosomal interactions are a mixture of true chromatin

interactions and interactions that are the result of random ligations (<https://doi.org/10.1038/s41592-021-01248-7>)

We generated a total of 419 million HiC-reads, of which 282 million reads were having ligation motifs and 169 million reads (HiC contact) were associated with DNA interaction. Indeed, to check HiC library quality, we checked *cis* and *trans* interactions as kindly suggested by the reviewer. We observed a high number of 96,089,039 *cis* (intra-chromosomal) interactions compared to *trans* (inter-chromosomal) 73,711,869 interactions, consistent with parameters indicating a good quality of the library (high *cis* interactions). This information is included in the supplementary table 2.

2. Figure 4A the TAD score and the triangle presented does not fit with each other?

The image is the direct output of the HiC explorer. We have improved the visualization of the figure to show alignment between TAD scores and triangles.

3. Juicer and Hic explorer, which are two different methods for analysis of the data, showed very different results and small overlap why?

Please see response to point 1 by reviewer 1 above.

4. To investigate whether local chromatin architecture remained conserved within legumes, authors selected *V. mungo*, *P. vulgaris*, *G. max* and *G. soja*. How can we compare data of different quality and resolution? This needs to be strongly justified as now is a major weakness of the comparative genomics approach. Furthermore, it is not clear at all that HiCexplorer is the best tool for that. See: <https://doi.org/10.1186/s13059-018-1596-9>

We would like to clarify here that we have re-analyzed the published data of *G. max*, *P. vulgaris* and *G. soja* with same resolution of a10kb as we did for *V. mungo*. This re-analysis was performed with the same intent as the reviewer has kindly pointed out- to have the same resolution as well as quality for our comparative analysis.

Hi-C explorer was adopted based on its use for a similar comparison used based in pepper performed more recently (Liao et al, 2022, Nature Communication, <https://www.nature.com/articles/s41467-022-31112-x>).

5. Authors wrote that " To gain insight into the genome wide DNA methylation landscape as well as its relationship with gene expression in *V. mungo*, we generated DNA methylome profiles of leaf, shoot and root tissues in two biological replicates at single base resolution by using whole-genome bisulfite sequencing (WGBS)". Expected correlations are described, but the novelty of the data in this section should be justified (just anticorrelation between DNA methylation and expression is not sufficient....)

This study reports, for the first time, publicly available resource of tissue level characterization of DNA methylation patterns in *Vigna mungo*. We have generated a whole genome landscape of *Vigna mungo* that had not been reported till now. We also show patterns of degrees of variation and conservation of DNA methylation among legumes (Figure 6). Therefore, these novel aspects of our data will serve as valuable tools for further in-depth DNA methylation studies in legumes.

6. Authors proposed that "low and intermediate levels of methylation might enhance transcription by increasing splicing efficiency of the primary transcripts." This is only one possibility over many others and more thorough analysis of splicing must be done (e.g. nascent RNAs to detect primary transcripts in relation with DNA methylation? Other marks?). What about histone modifications?

The focus of this study is on the genome-wide overview of comparative genomics and epigenomic aspects rather than the molecular level analysis. Analysis of nascent RNAs will require advanced sequencing experiments such as GRO-seq, FLEP sequencing etc. Similarly, ChIP-sequencing is required for investigating Histone modifications. These are currently out of the scope of this study and will be important aspects of future investigations.

Reviewer 3

This manuscript reports assembly of a high quality, near-complete, chromosome-level reference genome of *Vigna mungo*, and its spatial 3D genome organization (Hi-C) and global DNA methylation pattern. In addition, based on these data and those from the public repository, authors conducted comparative analyses from an evolutionary perspective in legume. Results of this study provide new genomics landscapes and 3D structures of *V. mungo*, which are useful in further studies of legume plants and crops. This study was well designed and executed; the manuscript overall was well written. I am not fully confident with some detailed analyzing tools in the Hi-C part due to unfamiliarity, nevertheless, I assume these are reliable given similar patterns (but see following comments) were reported in other plants, especially legumes, such as soybean. Apart from this less certain aspect, all conclusions are fairly supported by data.

We thank the reviewer for positive comments.

I have some additional concerns and suggestions, especially concerning Figures and Figure legends.

1. Although the writing is generally good and comprehensible, there are numerous grammar mistakes and inappropriate use of terminologies, for example, "transposons elements". Careful English-editing and proof-reading must be conducted before acceptance of the manuscript.

We thank the reviewer for this comment. We have worked on correcting grammatical mistakes throughout the text.

2. It was shown in *Glycine max* (Wang et al. 2021a) that for most chromosomes the A compartments tended to distribute at terminal regions, while B compartments trended towards the centromere. Author discussed this point in lines 456-460, but did not explain why *V. mungo* is so different as shown in Figure 1B.

We would like to clarify that our results are consistent with patterns observed in *G.max*, i.e, A compartment mainly associated with euchromatin regions and was located at chromosomal arms, while the B compartment mainly contained heterochromatin and was located at centromeric and pericentromeric regions of chromosomes. These findings are similar to what have been reported in *G. max*.

3. The color key of heatmap in Figure 1B is missing.

Heat map keys for the Circos plot are shown below the plot in Figure 1.

4. Line 122: I cannot find (H) in Figure 1B. Does it refer to the green line? Please clarify.

We thank the reviewer for pointing this out. Yes, the green lines show the syntenic relationship. We have updated the figure.

5. Given comparative genomics constitutes a major content of the paper, the evolutionary relationships among *V. mungo*, *V. angularis*, *V. unguiculata* *G. max* are better to be briefly described in the Introduction, as not all readers are familiar with the taxonomy of legumes.

We have added a few sentences related this in the introduction. (Lines 74-79)

6. Sources of the genome data of *V. angularis*, *V. unguiculata*, and *G. max* should be given in M&M.

We have cited the papers where these data were published. We have now included the SRA ID in the material and methods section.

7. Lines 170: 58 MYA is based on the current study or cited from a prior reference? Please clarify.

This information is published (Yang et al, 2015, PNAS). We have cited this reference in line 213

8. Please clarify the *V. mungo* and *V. angularis* in Figure 2C.

We have annotated the figure to specify *V. angularis* (top) and *V. mungo* (bottom)

9. Lines 204-206: These results seem different from that of Pootakham et al. 2021. Why only focused on *V. mungo* and *V. angularis*?

Although *V. radiata* is closely related to *V. mungo*, we did not use *V. radiata* genome because we found that the reported *V. radiata* genome is fragmented and therefore not suitable for this analysis (Yang et al 2014 Nature Communications). We therefore selected the next closely related species-*V. angularis* for our analysis because of its high-quality chromosomal-level assembly.

10. Lines 218-219: The data and figure do not match. Something went wrong in your description and/or Figure 2D?

We thank the reviewer for pointing this out. We have corrected the values for *V. mungo* in Figure 2D

11. Typo in Figure 2E. Please correct "Clade V".

We have corrected that.

12. Line 238: Maybe change Figure 2F to a line chart or density plot. It is difficult to get the important information about "0.5 to 0.8 MYA" from the current Figure 2F.

We thank the reviewer for this comment. We have now provided a density plot for clarity.

13. Line 423: What does 45D mean? 45 days of leaf. There is only one replicate of 45D leaf in Table S3?

Yes, it indicates leaf age. We have changed the sentence to explain this better- "The normalized expression level of each gene category was assessed using leaf samples (age 45 days after germination) RNA-seq data. We have updated the Table S3 with replicates.

14. Lines 455-456, 471-472, and 480-482: References are needed.

We have added the corresponding references.

15. Please uniform "Mb" or "mb", "Gb" or "GB", "HiC" or "Hi-C".

We have made these terms consistent throughout the text.

September 26, 2023

RE: Life Science Alliance Manuscript #LSA-2023-02074-TR

Dr. Sabhyata Bhatia
National Institute of Plant Genome Research
Aruna Asaf Ali Marg
New Delhi 110067
India

Dear Dr. Bhatia,

Thank you for submitting your revised manuscript entitled "Evolutionary insights into 3D genome organization and epigenetic landscape of *V. mungo*". We would be happy to publish your paper in Life Science Alliance pending final revisions necessary to meet our formatting guidelines.

- please address Reviewer 1's remaining comment
- please upload all figure files as individual ones, including the supplementary figure files; all figure legends should only appear in the main manuscript file
- please add ORCID ID for the corresponding author--you should have received instructions on how to do so
- please add a Summary Blurb/Alternate Abstract to our system
- please add the Twitter handle of your host institute/organization as well as your own or/and one of the authors in our system
- titles in the system and in the manuscript file must match
- please remove the figures from the manuscript text
- please add your main, supplementary figure, and table legends to the main manuscript text after the references section
- please add a conflict of interest statement to your main manuscript text
- there is only one panel in figures S1, S2, S4, S5 please remove A from the legends and actual figures
- please add a callout for Figure S6A-C to your main manuscript text
- please include a Data Availability statement at the end of the Materials and Methods section to indicate accession information for the sequencing data generated in this study
- the Author Contribution selected for Baljinder Singh does not qualify for authorship according to ICMJE guidelines. please either update this information or let us know if an author should be removed

A. FINAL FILES:

B. MANUSCRIPT ORGANIZATION AND FORMATTING:

Sincerely,

Reviewer #1 (Comments to the Authors (Required)):

Regarding the point #2, if the chromatin interactions (TLRs) callings are not reproducible, how the authors concluded low conservation among legumes?

Reviewer #3 (Comments to the Authors (Required)):

After a careful assessment of authors' responses and changes made in the revised manuscript, I consider authors have properly addressed all my concerns and comments. As said in my previous comments, I am not very familiar with the analyzing tools for Hi-C data and I am pleased to see these uncertainties have been pointed out by the other reviewers. As far as I am concerned, I judge this is an important study with clear novelty. The revised manuscript has been further improved.

Response to the remaining comment by Reviewer 1

Junaid et al, LSA-2023-02074-T

Reviewer 1

Regarding the point #2, if the chromatin interactions (TLRs) callings are not reproducible, how the authors concluded low conservation among legumes?

To identify conserved TAD features, a region with minimum overlap between two species is required. For example, in this study, we have defined conserved TADS where a 25% minimum ratio of bases (-minMatch = 0.25) for body and one-third for boundary (-minMatch = 0.33) should be overlapped. However, when we compare TADs features between biological or technical replicates within an organism, almost exact or at least 70-80% coordinates overlap or loci-by-loci comparison is used.

Reviewer #3

After a careful assessment of authors' responses and changes made in the revised manuscript, I consider authors have properly addressed all my concerns and comments. As said in my previous comments, I am not very familiar with the analyzing tools for Hi-C data and I am pleased to see these uncertainties have been pointed out by the other reviewers. As far as I am concerned, I judge this is an important study with clear novelty. The revised manuscript has been further improved.

We thank the reviewer for these positive and encouraging comments.

October 19, 2023

RE: Life Science Alliance Manuscript #LSA-2023-02074-TRR

Dr. Sabhyata Bhatia
National Institute of Plant Genome Research
Aruna Asaf Ali Marg
New Delhi 110067
India

Dear Dr. Bhatia,

Thank you for submitting your Research Article entitled "Evolutionary insights into 3D genome organization and epigenetic landscape of *V. mungo*". It is a pleasure to let you know that your manuscript is now accepted for publication in Life Science Alliance. Congratulations on this interesting work.

DISTRIBUTION OF MATERIALS:

Again, congratulations on a very nice paper. I hope you found the review process to be constructive and are pleased with how the manuscript was handled editorially. We look forward to future exciting submissions from your lab.

Sincerely,
